# Optimal Wavelength Selection for Hyperspectral Imaging Evaluation on Vegetable Soybean Moisture Content during Drying

**Peng Yu [1], Min Huang [1],\*, Min Zhang [2] and Bao Yang [3]**

[1] Key Laboratory of Advanced Process Control for Light Industry, Ministry of Education, Jiangnan University, Wuxi 214122, China; yupeng091333@163.com

[2] State Key Laboratory of Food Science and Technology, Jiangnan University, Wuxi 214122, China; min@jiangnan.edu.cn

[3] Department of Electrical and Computer Engineering, Oakland University, Rochester, MI 48309, USA; byang2345@oakland.edu

\* Correspondence: huangmzqb@163.com

**Abstract:** Hyperspectral imaging technology is a promising technique for nondestructive quality evaluation of dried products. In order to realize real-time, online inspection of quality of dried products, it is necessary to determine a few important wavelengths from hyperspectral images for developing a multispectral imaging system. This study presents a binary firework algorithm (BFWA) for selecting the optimal wavelengths from hyperspectral images for moisture evaluation of dried soybean. Hyperspectral images over the spectral region 400–1000 nm, were acquired for 270 dried soybean samples, and mean reflectance was calculated from hyperspectral images for each wavelength. After selecting 12 important wavelengths using BFWA, a moisture prediction model was developed using partial least squares regression (PLSR). The PLSR model with BFWA achieved a prediction accuracy of $R_p = 0.966$ and $RMSEP = 5.105\%$, which is better than those of successive projections algorithm ($R_p = 0.932$ and $RMSEP = 7.329\%$), and the uninformative viable elimination algorithm ($R_p = 0.928$ and $RMSEP = 7.416\%$). The results obtained by BFWA were more stable, with a smaller standard deviation of $R_p$ and $RMSEP$ than those of the genetic algorithm. The BFWA method provides an effective mean for optimal wavelength selection to predict the quality of soybeans during drying.

**Keywords:** hyperspectral imaging; wavelength selection; vegetable soybean; binary firework algorithm; PLSR

## 1. Introduction

As a popular snack food, dried soybeans are preferred by consumers because they are easy to carry and eat. Moisture content is an important parameter used in evaluating the drying quality of dried soybeans [1]. Traditional methods for moisture content measurement, such as the gravimetric oven method and the Karl Fisher titration, destroy dried samples in the testing process. The mentioned conventional measurement methods require considerable time to measure a large number of samples, which is time-consuming and laborious [2]. Considering the limitations of traditional methods, several studies have focused on the development of nondestructive technologies in evaluating the moisture content of dried samples. Machine vision and near-infrared spectroscopy are the two main methods for rapid nondestructive measurement [1,3,4]. These two methods have several limitations, even if they can rapidly detect the sample dryness without destroying samples. The traditional machine vision method can only obtain a limited detection range in detecting samples. The complete average image

information cannot be obtained due to visual range limitation, which leads to inaccurate measurement. Near-infrared radiation (NIR) covers the range of the electromagnetic spectrum between 780 and 2500 nm. Since water has several strong absorption wavelengths at this wavelength range, NIR spectroscopy is widely used in moisture measurement of various agriculture products. However, it cannot obtain the spatial information of samples. Thus, the results obtained by these two methods may lead to the loss of valuable information. Hyperspectral imaging possesses the advantages of the two methods and overcomes their shortcomings via rapid nondestructive measurement. Hyperspectral imaging can provide detailed or complete information, such as chemical composition, internal structure characteristics, and morphological information [5]. Thus, this technology may be used as an alternative method for predicting the moisture content of vegetable soybeans during drying.

In the process of predicting moisture content of vegetable soybeans by using a hyperspectral system, not all spectral wavelengths are beneficial for wavelength modeling, because many spectral data exist in the entire spectrum. Adjacent wavelengths may have high correlation and redundancy that increase the storage space of data. The actual calculation amount is remarkably increased, and the accuracy and real-time of the result are directly affected. Therefore, the selection of optimal wavelengths is important for rapid and accurate non-destructive analysis of agricultural products using hyperspectral images. Meanwhile, a reduction of the dimensions of hyperspectral image data reduces the impact on the overall recognition accuracy.

Various algorithms have been applied to select optimal wavelength, such as the successive projections algorithm (SPA), which has been used to eliminate redundant information in near-infrared spectroscopy in order to analyze the moisture content of wheat [6]. An uninformative viable elimination algorithm (UVE) has also been applied in wavelength selection to establish an apple firmness prediction model [7,8], and a genetic algorithm (GA) was used for wavelength selection in remote sensing hyperspectral systems to improve the observation of surface objects [9,10]. Although these algorithms have been applied for optimal wavelength selection, the accuracy of the results cannot be guaranteed by using these methods in trying to detect the moisture content of vegetable soybeans.

A firework algorithm (FWA) [11] imitates the actual process of a firework explosion. A firework explosion in this context is defined as the process of individual firework that automatically searches for the optimal solution in the solution area. The FWA method has been applied to many practical problems in recent years. FWA was used to optimize a radial distribution network, solving a network reconfiguration aimed at improving power loss minimization and the voltage profile of a distribution system [12]. FWA was applied to image segmentation by using an adaptive transfer function. The experimental results demonstrated that the FWA method performed better than other methods with regards to segmentation for most of the images [13]. As a variant of FWA, the binary firework algorithm (BFWA) was achieved by FWA coupled with a binary string ("1" and "0") [14]. This particular operation makes BFWA potentially useful for selecting the optimal wavelengths from hyperspectral imaging data. The specific objectives are expressed as follows:

- Optimal wavelengths based on the mean reflectance value of dried vegetable soybeans are selected by using the BFWA.
- A partial least squares regression (PLSR) method is used to develop prediction models for predicting the moisture content of dried soybeans.

## 2. Materials and Methods

### 2.1. Experimental Samples and Data Acquisition

Vegetable soybeans used in this experiment were purchased from the vegetable soybean planting base of the Zhejiang Cixi Haitong Food Co., Ltd. Fresh vegetable soybeans should be pretreated. After cleaning, peeling, and blanching, the fresh vegetable soybeans were stored in cold storage at a temperature of 4 °C and 95% humidity in preparation for the next experiment, in which the preservation time should not exceed 3 days. In this experiment, about 200 g of vegetable soybeans

were dried by using pulse spray microwave vacuum drying equipment. The drying times were set to 0–80 min with steps of 10 min (here, 0 min means the soybeans are not dried). The dried soybeans were randomly divided into 10 portions (about 5 soybeans for each portion), and each portion was treated as one sample for image collection and moisture measurement. Each drying time experiment was repeated thrice to eliminate unnecessary measurement errors. Hence, a total of 270 (10 samples, 3 repeats, 9 drying times) samples were obtained after drying. In this study, moisture content measurement was conducted by using a convection oven in reference to the National Standard of China (GB/T/8858-88).

An in-house line-scan hyperspectral imaging system was used to acquire the hyperspectral reflectance images of the soybean samples. This system consists of a single optic fiber coupled with a 9 inch parallel light powered by a 150 W DC light source (EKE, 3250K, TechniQuip, San Francisco, CA, USA), a back-illuminated 1392 × 1024 pixel charge-coupled device (CCD) camera (Pixelfly QE IC/285AL, Cooke, USA), an imaging spectrograph (1003A-10140 Hyperspc™ VNIR C-Series, Head-wall Photonics Inc., Fitchburg, MA, USA) with a 20 μm slit covering an effective range of 400–1000 nm connected with a zoom lens (10004A-21226 Lens, F/1.4 FL23 mm, Standard Barrel, C-Mount., USA), a sample delivery platform driven by a stepping motor, and a computer supported with camera control and image acquisition. For image acquisition, 10 soybeans as one group were placed onto a 20 cm × 20 cm black backgrounded board in two rows. For each group of soybeans, 625 scans covering a 50 mm distance were acquired by 80 μm scanning steps at an exposure time of 250 ms. After 10 spectral binning operations, a 1392 × 625 × 94 image spatial block was created, which had 6.44 nm/pixel spectral resolution, 0.15 mm/pixel spatial resolution, and 94 wavelengths.

### 2.2. Image Preprocessing, Segmentation, and Feature Extraction

For reducing the light source effect, darkness and reflectance images of white Teflon were also collected for use as a reference to correct the raw reflectance image of each sample. The relative reflectance image was calculated by dividing the difference in energy readings between sample and darkness by the difference in energy readings between the white Teflon and darkness. All the relative reflectance images were used in following analyses.

In order to extract useful information for model developing and testing, segmentation of the vegetable soybeans from the hyperspectral image background is a key step. Threshold segmentation was used, because of convenient calculation and high accuracy. In this study, the reflectance image of soybean samples at 718.2 nm was selected, because of its superior contrast outline in this image (step I) [1]. Prior to image segmentation, the obtained gray image had noise removed by image filtering and enhancement (step II). Threshold segmentation was then performed to produce a binary mask by expansion, corrosion, opening, and closing of the gray image (step III). The obtained binary mask was used to segment the true soybean regions from the 94 wavelengths in the image spatial block (step IV). Finally, the mean relative reflectance of the 270 soybean samples was calculated after the automatic segmentation of the image background, and each soybean from the true image yielded large amounts of spatial and spectral information. Considering the physical state and chemical composition changes of the soybeans during the drying process, mean reflectance was applied to extract and predict the moisture content characteristics of the dried soybeans. The mean reflectance of each dried soybean was calculated in following equation:

$$Mean = \frac{1}{M_x * M_y} \sum_{i=1}^{M_x} \sum_{j=1}^{M_y} f(i,j), \tag{1}$$

where $f(i,j)$ is the relative reflected light intensity of each pixel. $M_x$ and $M_y$ are the number of pixels in the horizontal and vertical directions of the sample, respectively.

*2.3. Wavelength Selection Based on BFWA*

2.3.1. SPA, UVE and GA

Each of extracted spectra consists of 94 wavelengths, which contain redundant information between adjacent wavelengths. Selecting the optimal wavelengths for developing more robust prediction models and for implementing a multispectral imaging system is desirable. SPA and UVE are two traditional methods used for wavelength selection. SPA is a forward selection method and aims to solve the co-linearity problem with minimal redundancy, based on its principle of establishing the candidate subsets of the most representative variables according to a sequence of projection operations relating the columns of the instrumental response matrix [15]. This method starts from a wavelength. Then, the wavelength is cyclically projected onto an unselected wavelength to incorporate a new one at each iteration. Each wavelength containing more characteristic information will be included in the optimal wavelength set until a specified number of wavelengths are reached. UVE is a method of variable selection based on a stability analysis of the regression coefficient. UVE can eliminate the wavelength variables, which make less of a contribution to the prediction ability of the model. Employing the variables selected by UVE for modeling can avoid a model over-fitting, and usually improves its predictive ability [16]. As a traditional and widely-used group intelligent algorithm, GA applies a "survival of the fittest" approach for wavelength selection. GA is based on the natural selection and genetic mechanisms of the biological world and uses selection operators, exchange, and mutation to achieve the best solution [10]. In this study, SPA, UVE and GA algorithms were used for optimal wavelength selection to compare the model's performance with the proposed BFWA algorithm.

2.3.2. BFWA

As a swarm intelligence algorithm, FWA is inspired by actual firework explosions that illuminate the surrounding area, and has the advantages of a search, explosiveness, and diversity in parallel. Conventional FWA is suitable for searching for an optimal value of a continuous system, but not suitable for discrete problems, because of the coding mode. To overcome the disadvantage of conventional FWA, a binary firework algorithm (BFWA) was proposed. The BFWA was achieved by FWA combined with binary coding in solving a discretization problem, and can be applied for the selection of optimal wavelengths. In this study, the location of fireworks in FWA was encoded. Similarly to the FWA implementation process, population initialization, evaluation, explosion, generation of "sparks", and selection operators were used based on fitness in the BFWA.

Population Initialization

The obvious difference between the BFWA and original FWA is that the firework location is defined as a binary string. In the BFWA method, binary numbers "1" and "0" are used to indicate whether the hyperspectral image information at a certain wavelength is selected ("1" indicates that these wavelengths are selected in the evolutionary process, and "0" indicates that these wavelengths are excluded) [14,17]. The remaining number of wavelengths after selection is $m$, and $N$ firework locations are distributed in the analytic domain to randomly combine the initial population in initializing the original firework population. First, a number of (0, 1) is randomly generated at each location of the 94 wavelength locations. Then, the obtained random array is sorted in order from large to small, and the previous $m$ values are selected to record the corresponding wavelength location. The wavelength location that corresponds to the largest number of the first $m$ is placed in "1," and the remaining $(94 - m)$ wavelength locations are "0." This process is repeated $N$ times, in which an initial population that contains $N$ fireworks can be obtained. In this study, the number of fireworks ($N$) in the population was set to 20.

Evaluation

In this experiment, a PLSR model [18] and cross-validation were used for evaluation. The PLSR model was established by using the mean reflectance feature obtained from the selected wavelengths. The number of potential variables of the model was determined by using a leave-one-out validation method, coupled with PLSR. The value of the root mean square error of cross-validation was used as the fitness value of the selected fireworks or sparks. The smaller the fitness value is, the better the quality of firework or spark locations will be.

Explosion

The "explosion" is the key operator of BFWA, which includes the number of sparks and explosion amplitude as two important parameters. The number of sparks during an explosion is used to determine the new firework in the evolution process. The number of sparks is expressed as:

$$num_i \ = \ Num * \frac{Y_{\max} - f(X_i) + \varepsilon}{\sum\limits_{i\,=\,1}^{N}\left(Y_{\max} - f(X_i)\right) + \varepsilon}, \tag{2}$$

where $num_i$ is the number of sparks produced by firework $X_i$, and $Num$ is a constant, which is used to control the number of sparks generated by each firework. Generally, a smaller value of $Num$ makes it difficult for the BFWA to find the optimal wavelengths, and larger $Num$ will bring unnecessary computation and affect the evolution speed. In this study, $Num$ was chosen to be 400, after balancing between a global optimal solution and evolution time through preliminary experiments. $N$ is the number of fireworks in the population, and $f(X_i)$ is used to calculate the fitness value function. $Y_{\max}$ is the maximum fitness value in the population. To avoid a zero denominator, a minimum constant $\varepsilon$ is added to the numerator and denominator. $\varepsilon$ was set to $1e - 38$ [19].

To avoid excessive or few sparks during the operation, the number of sparks generated is limited by using Formula (3):

$$S_i \ = \ \begin{cases} S_{\min} & if\ num_i < S_{\min} \\ S_{\max} & if\ num_i > S_{\max} \\ round(num_i) & otherwise \end{cases}, \tag{3}$$

where $S_i$ is the actual number of sparks produced by firework $X_i$, $num_i$ is the number of sparks calculated based on Formula (2), $S_{\min}$ is the least number of fireworks in the explosion process, and $S_{\max}$ is the maximum number of sparks produced by each firework. Through many simulation and data analyses, when the number of sparks produced by fireworks was too large, the amount of calculation was increased. If the number of sparks was too small, however, it will produce an incomplete search in the local analytical domain. Considering all facets comprehensively, $S_{\min}$ was set to be 10 and $S_{\max}$ was set to be 40 by trial and error method to find the optimal solution.

Explosion amplitude refers to the scope of finding other solutions near a firework, which can control the range of an explosion in the position of fireworks [11]. Compared to the original FWA, in the BFWA explosion amplitude is replaced by explosion radius. The range of finding the best solution is accurate (explosion radius becomes smaller) with the increase of iterations, and the definition of explosion radius $A_i$ is shown as Formula (4) [20]:

$$A_i \ = \ \left(\frac{T - t}{T}\right)^{expo}\left(A_{init} - A_{end}\right) + A_{end}, \tag{4}$$

where $A_i$ is the explosion radius of firework $X_i$, $T$ is the maximum number of iterations, $t$ indicates the number of current iterations, and $A_{init}$ is the maximum radius of the initial firework explosion. $A_{end}$ is the largest explosion radius of the last firework explosion. $expo$ is a constant. Considering the complexity and running time of the BFWA, the parameters were selected by some preliminary

experiments and related references. We found that the BFWA worked quite well at the setting: $A_{init}$ = 5, $A_{end}$ = 50, $expo$ = 0.04 [21].

Generation of Sparks and Selection

The key step in the selection of optimal wavelengths by BFWA is the generation of new locations. The location of a spark is based on the location of fireworks. The spark location contains a lot of information about fireworks. The spark location can obtain much information that several original fireworks do not have by changing the operation. A total of 94 wavelength positions are used for the selective operation. The steps for generating sparks from a firework $X_i$ are as follows:

Step 1: $A_i$ is the explosion radius of firework $X_i$, which is calculated from Formula (4). $u$ is used as the starting point for the change of firework position, in which an integer is randomly selected from $(0, 94 - A_i)$.

Step 2: An explosion occurs at $(u, u + A_i)$ location. For the wavelength position of the explosion interval, each of the wavelength locations add a random number from (0, 1) to the original location.

Step 3: A new random number set is arranged from large to small. For this arrangement order, the wavelength location that corresponds to the largest number of the first $m$ is placed in "1," and the remaining $(94 - m)$ wavelength locations are "0." Each firework generates a certain amount of sparks based on Formulas (2) and (3). The updated rule of mutation location is shown in Figure 1.

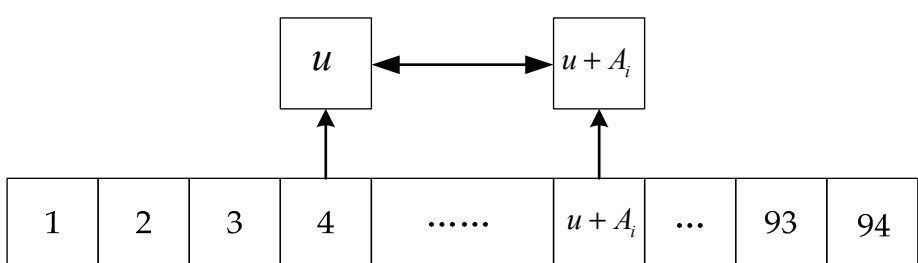

**Figure 1.** Explosion diagram of the firework $X_i$.

Step 4: Decode the spark locations and calculate the fitness of sparks through the fitness function. In order to keep the population size unchanged and the diversity of sparks in the evolution process, the best $N/2$ fitness values from all fireworks and sparks are first selected, and the other $N/2$ locations are randomly selected from the remaining fireworks and sparks. The two parts are made for initial $N$ firework locations of the next iteration.

Optimal wavelengths were determined based on the smallest fitness value after the termination of iterations. Because BFWA is a random search algorithm, the maximum iteration number of BFWA was fifty to reduce the randomness effect in the process. This method provides a possible way to make fireworks to produce appropriate sparks and effectively improve global search capability. Sparks can retain most information from the original fireworks and can add several pieces of information that the original fireworks do not have. The number of selected wavelengths can be controlled by this manner of explosion. A rough framework of the BFWA is depicted in Figure 2.

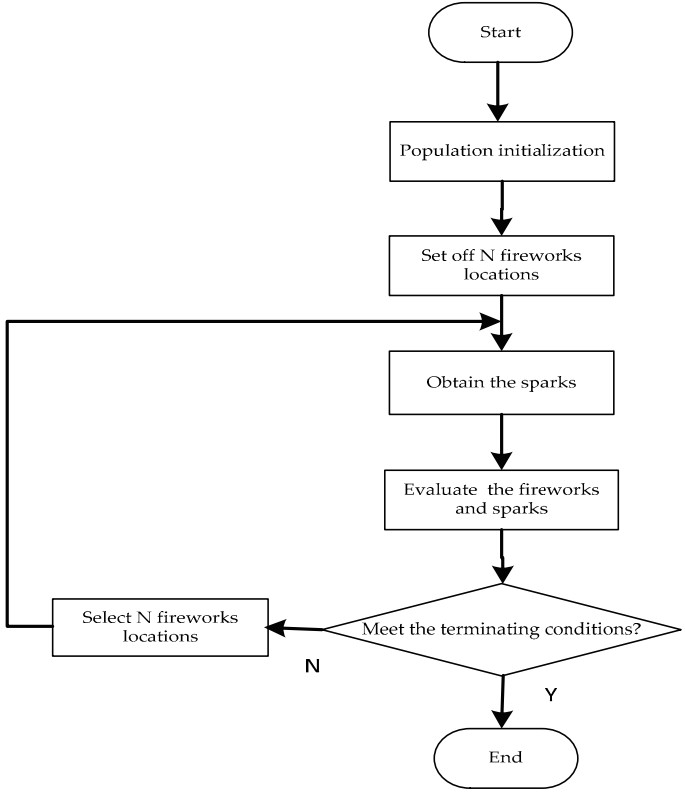

**Figure 2.** Framework of the binary firework algorithm (BFWA).

### 2.4. Development of a Moisture Prediction Model of Dried Vegetable Soybeans

PLSR was used to predict the moisture content of dried vegetable soybeans using the optimal wavelengths. Prior to model development, samples were randomly divided into two sets with 75% for calibration and the remaining 25% for independent prediction. The PLSR model coupled with leave-one-out cross validation was then applied to develop the calibration model using the optimal wavelengths. The prediction model was validated with a prediction set based on the same wavelengths, in terms of the correlation coefficient and standard error between predicted and the measured moisture content. The values of the related coefficient and root mean square error were used as evaluation indicators of the models. In the experiment, to better compare the performance of the models, the calibration and prediction procedures were repeated 10 times to ensure that the reported results more accurately reflected the actual performance of the modeling method. The average values of correlation coefficients (i.e., $R_c$ and $R_p$) and root mean square error for calibration and prediction (i.e., *RMSEC* and *RMSEP*) from 10 runs were used to evaluate the performance of the model [1,22].

All the algorithms were performed in MATLAB R2014b (MathWorks Inc., USA), with a PLS-toolbox 5.0 (Eigenvector Research, Inc., Wenatchee, WA, USA).

## 3. Results and Discussion

### 3.1. Prediction Model Established by Full Wavelengths

Figure 3 shows the average of the reflectance spectra over one portion (five soybeans) for processed samples at different drying times (0, 10, 40, 50, and 70 min). Absorption peaks around 430, 535, 660, and 970 nm were observed, which were caused by carotenoid, anthocyanin, chlorophyll, and moisture contained in the vegetable soybeans [1]. At the initial periods of drying (0, 10, and 40 min), changes of color and microstructure occurred on the surface, caused by evaporation of water in the immediate surroundings of the soybean surface. These are the main factors affecting the reflectance spectra. In these periods, the discoloration and microstructural change gradually extended from slight to

moderate, which led to the decrease of the reflectance spectra. At a later stage of drying (50 and 70 min), the outer layer of soybean cells on the surface became "unsaturated" with moisture, and the diameter of pores and capillaries decreased sharply, resulting in increase of shrinkage and densification of the surface microstructure. This compact and brittle structure of soybeans may account for the increase in reflectance spectral value observed at 50 and 70 min of drying [1].

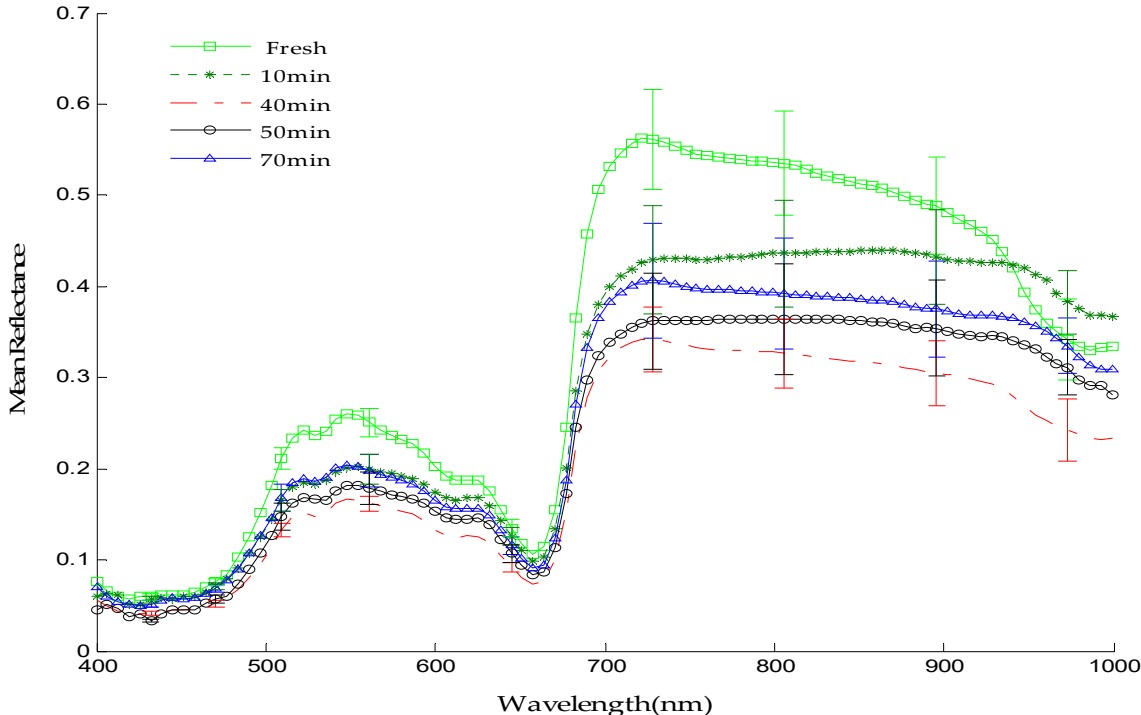

**Figure 3.** Mean reflectance spectra of five soybeans at different drying times (0, 10, 40, 50, and 70 min).

Before selecting the optimal wavelengths, the PLSR model using the full wavelength spectra of soybean was built to predict the moisture content of the dried soybeans. After 10 runs, the average values of $R_c$, RMSEC, $R_p$, and $RMSEP$ were 0.984, 3.523%, 0.971, and 4.733%, respectively. Although the models yielded good accuracy based on the full wavelength, it was difficult to realize online detection of moisture content, due to the speed limitations in imaging acquisition and processing.

### 3.2. Comparison of Results Obtained by SPA, UVE, and BFWA

Three different algorithms (SPA, UVE, and BFWA) were used in the wavelength selection, and a PLSR model was utilized to compare the predicted values. Figure 4 shows the results of the selection of different wavelength numbers by different methods. Compared with SPA and UVE, the BFWA achieved better prediction performance. The correlation coefficient $R_p$ values obtained by the BFWA was higher compared to the two other methods in selecting different numbers of wavelengths. Meanwhile, the *RMSEP* values were lower. The results showed that the BFWA performed better in predicting moisture content.

The accuracy and real-time of the results are important considerations in the development of online detection of sample moisture content systems. On this basis, the selection of optimal wavelengths is essential for reducing the amount of data acquisition and processing. As shown in Figure 4, the results of the model were stable when the number of selected wavelengths was 12. The selection of optimal wavelengths is an effective way to reduce the calculation time of prediction models and the amount of data processing. Considering the two aspects of accuracy and real-time results, 12 was a reasonable number of selected wavelengths. The results obtained by three different methods are shown in Table 1.

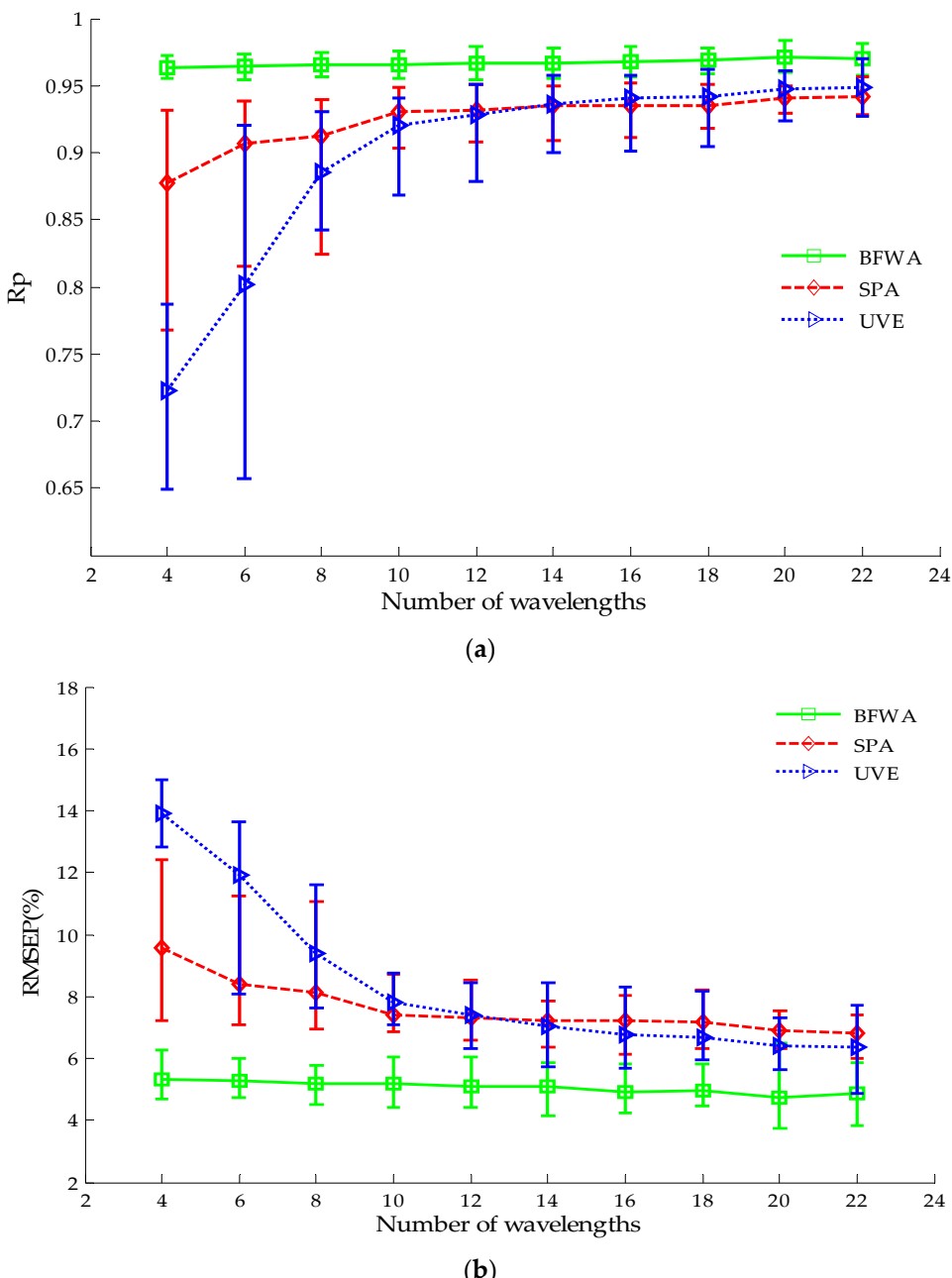

**Figure 4.** (**a**) The $R_p$ values obtained by successive projections algorithm (SPA), viable elimination algorithm (UVE), and BFWA; (**b**) The *RMSEP* values obtained by SPA, UVE, and BFWA.

**Table 1.** Average of 10 calibration and prediction results for moisture content by the partial least squares regression (PLSR) model with 12 optimal wavelengths selected.

| Method | $R_c$ [a] | Std [c] of $R_c$ [a] | RMSEC [b] (%) | Std [c] of RMSEC [b] (%) | $R_p$ [a] | Std [c] of $R_p$ [a] | RMSEP [b] (%) | Std [c] of RMSEP [b] (%) |
|---|---|---|---|---|---|---|---|---|
| SPA | 0.943 | 0.006 | 6.711 | 0.292 | 0.932 | 0.015 | 7.329 | 0.537 |
| UVE | 0.942 | 0.008 | 6.728 | 0.428 | 0.928 | 0.024 | 7.416 | 0.731 |
| BFWA | 0.976 | 0.002 | 4.381 | 0.154 | 0.966 | 0.009 | 5.105 | 0.455 |

[a] $R_c$, $R_p$: correlation coefficient of calibration set and prediction set, respectively. [b] *RMSEC*, *RMSEP*: root mean square error of calibration set and prediction set, respectively. [c] **Std**: standard deviation of data.

As shown in Table 1, the $R_c$, *RMSEC*, $R_p$, and *RMSEP* values obtained by the BFWA were better compared to SPA and UVE. For the calibration sets, the averages of $R_c$ values were higher by 3.5%

and 3.6%, and the averages of *RMSEC* values were reduced by 34.7% and 34.9%. For the prediction sets, the averages of $R_p$ values were higher by 3.6% and 4.1%, and the averages of *RMSEP* values were reduced by 30.3% and 31.2%. Compared to the other two methods, the standard deviation (Std) values of $R_c$, *RMSEC*, $R_p$, and *RMSEP* obtained by BFWA method were smaller than those obtained by the SPA and the UVE method, which indicated that the BFWA method performed better in stability convenience. Although the accuracy values of the BFWA model were slightly lower than that of the full wavelength model, the wavelength number is only 12.8% of full wavelength, which helps to develop online multispectral imaging systems. For online systems, fewer wavelengths will reduce the time of data acquisition, transformation, and processing to meet real-time requirements in application.

The 12 wavelengths were 457.96, 509.48, 567.44, 586.76, 612.52, 657.6, 696.24, 747.76, 773.52, 850.8, 908.76, and 934.52 nm of a selective operation by BFWA. These wavelengths were mostly centered around the absorption peaks shown in Figure 3's characteristic curves, which play an important role in soybean moisture prediction. Figure 5 shows a scatter plot for predicting the moisture content of soybeans from the prediction set. The diagram indicates that a good correlation exists between the actual value and predicted value of soybean moisture content.

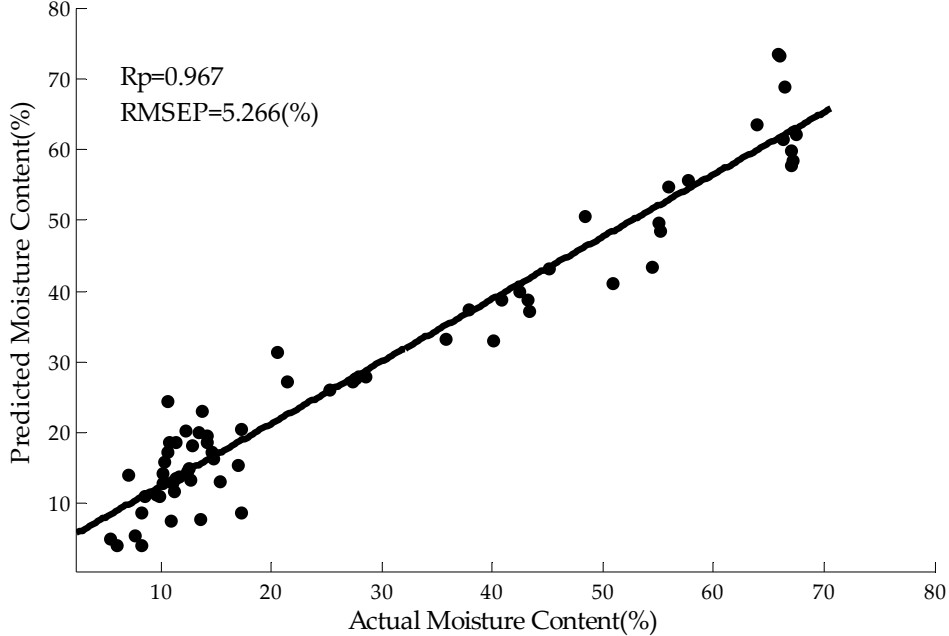

**Figure 5.** Moisture content prediction of soybeans using the PLSR model.

### 3.3. Comparison of Results Obtained by GA and BFWA

GA is an important swarm intelligent algorithm, and is widely used for feature wavelength selection in hyperspectral images. To further prove that BFWA can be used for wavelength selection, BFWA was compared to a GA algorithm for wavelength selection. Using a trial and error strategy, the optimal model parameters of GA were set as follows: population size was 30, maximum iteration number was 50, crossover probability was 0.6, and mutation probability was 0.01.

Figure 6 shows the results of the prediction model established by PLSR in selecting different number of wavelengths based on the two methods. Compared to GA, the average correlation coefficient $R_p$ and average *RMSEP* values of 10 prediction sets obtained by BFWA were improved at different number of wavelengths.

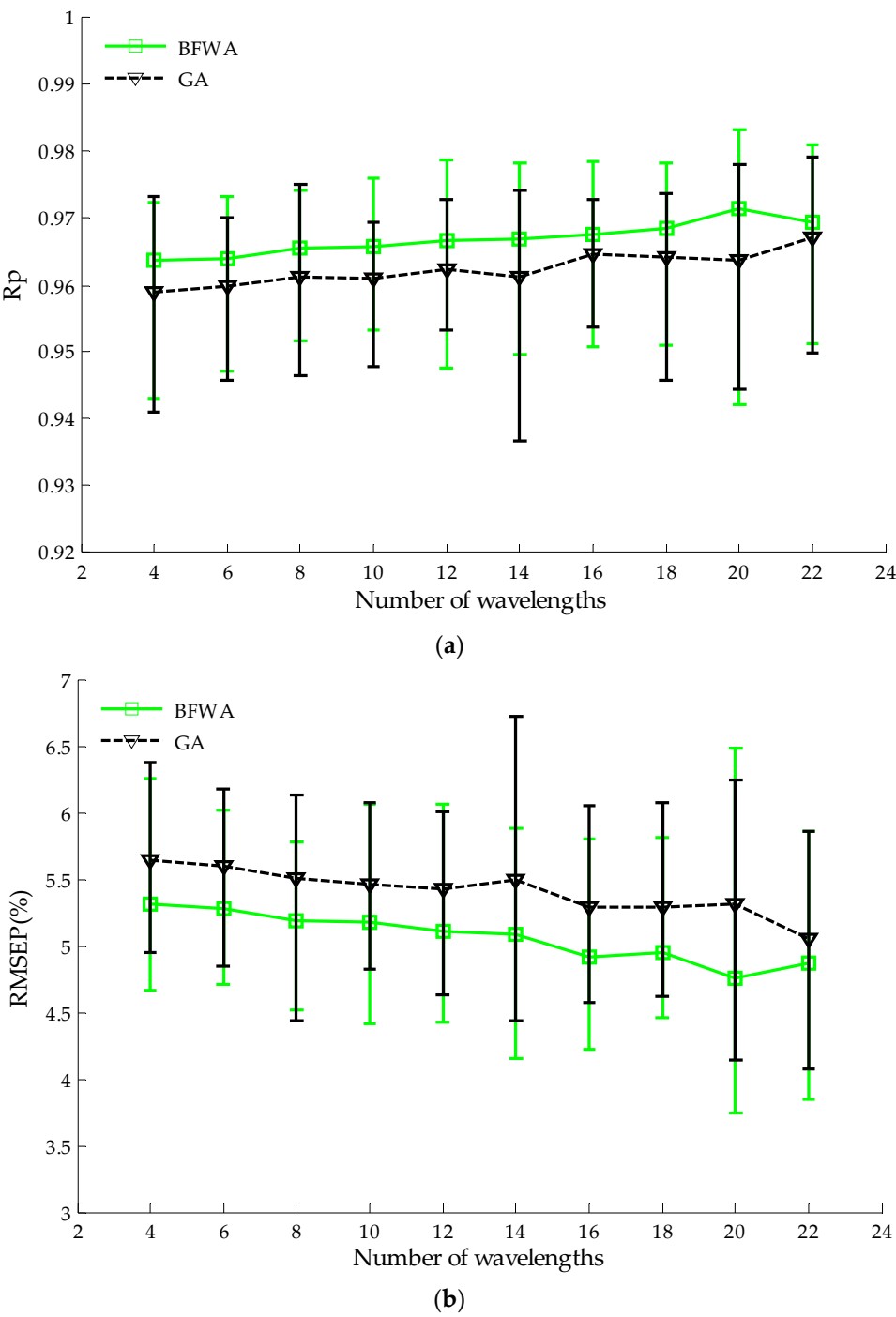

**Figure 6.** (**a**) The $R_p$ values obtained by genetic algorithm (GA) and BFWA; (**b**) The *RMSEP* values obtained by GA and BFWA.

As swarm intelligent algorithms, the values of initial population in GA or BFWA have an effect on the model performance. A good algorithm should be robust (insensitive) to the value changes of the initial population. That is, the accuracy obtained with different values for the initial population should be stable. In this study, each calibration and prediction set was repeated 10 times by setting different value of initial population. After 10 repeated runs for each prediction set, the mean values and Std of *RMSEP* and $R_P$ obtained by the PLSR model with 12 selected wavelengths, as shown in Table 2. The BFWA obtained slightly better prediction results (the average of $R_p$ and *RMSEP* are 0.967% and 5.074%) than the GA method (the average of $R_p$ and *RMSEP* are 0.962% and 5.439%).

Although there was only a small improvement in $RMSEP$ and $R_p$, the statistical test (*t*-test) shown that the performance of BFWA was statistically superior to that of GA ($p < 0.05$). Meanwhile, compared with GA, the Std values of $R_P$ and $RMSEP$ values were larger than that of BFWA, except for prediction sets 2 and 4, which indicates that the BFWA performed better than the GA in terms of stability.

**Table 2.** Stability results of the PLSR model obtained by the two methods with 12 optimal wavelengths selected.

| Prediction Set | Method | Mean Value of $R_p$ [a] | Mean Value of $RMSEP$ [b] (%) | Std [c] of $R_p$ [a] | Std [c] of $RMSEP$ [b] (%) |
|---|---|---|---|---|---|
| Prediction set 1 | GA | 0.965 | 5.440 | 0.004 | 0.298 |
| | BFWA | 0.970 | 5.127 | 0.003 | 0.205 |
| Prediction set 2 | GA | 0.948 | 5.987 | 0.002 | 0.122 |
| | BFWA | 0.949 | 5.913 | 0.003 | 0.161 |
| Prediction set 3 | GA | 0.951 | 5.546 | 0.005 | 0.242 |
| | BFWA | 0.962 | 4.949 | 0.003 | 0.203 |
| Prediction set 4 | GA | 0.968 | 5.364 | 0.002 | 0.171 |
| | BFWA | 0.976 | 4.704 | 0.003 | 0.250 |
| Prediction set 5 | GA | 0.960 | 5.666 | 0.003 | 0.239 |
| | BFWA | 0.965 | 5.274 | 0.002 | 0.159 |
| Prediction set 6 | GA | 0.960 | 5.536 | 0.004 | 0.233 |
| | BFWA | 0.964 | 5.122 | 0.002 | 0.183 |
| Prediction set 7 | GA | 0.970 | 5.211 | 0.003 | 0.203 |
| | BFWA | 0.972 | 4.972 | 0.002 | 0.136 |
| Prediction set 8 | GA | 0.962 | 5.388 | 0.001 | 0.057 |
| | BFWA | 0.972 | 4.782 | 0.001 | 0.038 |
| Prediction set 9 | GA | 0.970 | 4.884 | 0.003 | 0.237 |
| | BFWA | 0.973 | 4.641 | 0.001 | 0.105 |
| Prediction set 10 | GA | 0.961 | 5.374 | 0.004 | 0.227 |
| | BFWA | 0.966 | 5.254 | 0.001 | 0.107 |
| **Average** | GA | 0.962 | 5.439 | 0.003 | 0.203 |
| | BFWA | 0.967 | 5.074 | 0.002 | 0.155 |

[a] $R_p$: correlation coefficient of prediction set. [b] $RMSEP$: root mean square error of prediction set. [c] **Std**: standard deviation of data.

*3.4. Discussion*

Analyzing the prediction results obtained by using different wavelength selection methods (SPA, UVE, GA, and BFWA), BFWA had better prediction results than that obtained by the other two traditional wavelength selection methods (SPA and UVE). The accuracy of prediction results obtained by the GA method was slightly worse than the BFWA method. The reason for these differences is due to the fact that GA and BFWA belong to swarm intelligence algorithms, i.e., both algorithms generate an initial population randomly in the solution space. When the algorithms search for the optimal solution in the global solution space with continuous iterations, the search is more concentrated on high performance, which is conducive to finding the optimal solution quickly. This feature of swarm intelligence algorithms is more suitable for wavelength selection of hyperspectral images than traditional methods. There are some similarities among swarm intelligence algorithms in the iteration process, which may lead to the results of prediction models obtained by the BFWA and GA methods being close to each other. Real-time results are another important consideration in the development of online detection of sample moisture content systems. When the same prediction set was used to predict the moisture content, the values of average calculation time were about 0.531 and 0.296 seconds by using full wavelengths and 12 optimal wavelengths, respectively. It is noteworthy that some parameters need to be set in this algorithm, which have certain impacts on the wavelength selection.

We obtained a set of better parameter combinations through a large number of repeated experiments and data analysis. However, a better method to determine parameters needs to be further studied.

## 4. Conclusions

A binary firework algorithm (BFWA) was found to be suitable for selecting optimal wavelengths from hyperspectral images. When 12 optimal wavelengths were selected, the BFWA method ($R_p$ and $RMSEP$ are 0.966 and 5.105%) obtained the better prediction results than the SPA method ($R_p$ and $RMSEP$ are 0.932 and 7.329%) and the UVE method ($R_p$ and $RMSEP$ are 0.928 and 7.416%). Compared to the GA method, the results obtained by BFWA were more stable because of smaller Std of $R_p$ and Std of $RMSEP$. Furthermore, research on the performance of a BFWA on other crops or other species of soybeans is needed in the future.

**Author Contributions:** P.Y., M.H., M.Z. and B.Y. conceived and designed the experiments; M.H. performed the experiments; M.Z. and B.Y. analyzed the data; P.Y. and M.H. wrote the paper. All authors collaborated on the interpretation of the results and on the preparation of the manuscript.

**Funding:** This research was funded by the China Key Research Program (Contract No. 2017YFD0400901), the National Natural Science Foundation of China (Grant no. 61775086), the Fundamental Research Funds for the Central Universities (JUSRP51730A), and sponsored by the 111 Project (B12018).

**Acknowledgments:** Min Huang, Min Zhang and Peng Yu gratefully acknowledge the financial support from the China Key Research Program (Contract No. 2017YFD0400901), the National Natural Science Foundation of China (Grant no. 61775086), the Fundamental Research Funds for the Central Universities (JUSRP51730A), and sponsored by the 111 Project (B12018).

**Conflicts of Interest:** The authors declare no conflict of interest.

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
