# Peer review of "Optimal Wavelength Selection for Hyperspectral Imaging Evaluation on Vegetable Soybean Moisture Content during Drying"

_applsci, doi:10.3390/app9020331_

Round 1
Reviewer 1 Report
The manuscript has been improved somewhat but there are still issues remaining to be addressed:
1. The main question about this paper is that the results for the preferred technique (BFWA) gives averaged Rp and RMSEP of 0.966 and 5.1% respectively, which is so close to that of other methods, eg, the GA with Rp and RMSEP of 0.962 and 5.4% respectively. In practical term they are the same and it is not justified to make the conclusion ‘ The BFWA method performed better than the successive projections algorithm (SPA), uninformative viable elimination algorithm (UVE), and genetic algorithm (GA) in selecting different numbers of wavelengths.’ The results presented in fig 6 and table 2 could only address statements something like ’..the two results appear to be similar within experimental errors..’ . Particularly, the preferred technique (BFWA) was well tried and the results were obtained using ‘optimal’ model parameters. How does it perform when sub-optimal model parameters are employed? What about the other methods and are they tested using optimal model parameters too? The authors must address these issues and to make appropriate statements in the abstract and conclusion sections accordingly.
2. In my previous review the following questions have not been addressed in this version:
a. ‘section 3: Please check fig 3: is the legend not labelled correctly? How could the 40mins dried sample exhibits a lower norm that that of the 70mins dried spectra? Also, the figure shows a ‘mean’ spectra but the text didn’t mention whether this mean is the average of the spectra over several pixels of the same soybean or it is the average over several soybeans?’
b. ‘Table 2: please add a row of averaged figures (Rp, RMSEP, std etc) for GA vs BFWA over the 10 runs at the bottom of the table?’
c. ‘The conclusion in section 5 may be a bit over interpreted, furthermore, all results presented is based on one sample set of soybeans: it may be better (though not absolute essential) to include other species of soybeans or other crops to strengthen the argument that the BFWA performs more robust than all adopted technique!’
3. Use of English: there are quite a few grammatical/spelling mistakes throughout the manuscript that are needed to address.
Author Response
Point 1: The main question about this paper is that the results for the preferred technique (BFWA) gives averaged Rp and RMSEP of 0.966 and 5.1% respectively, which is so close to that of other methods, eg, the GA with Rp and RMSEP of 0.962 and 5.4% respectively. In practical term they are the same and it is not justified to make the conclusion ’The BFWA method performed better than the successive projections algorithm (SPA), uninformative viable elimination algorithm (UVE), and genetic algorithm (GA) in selecting different numbers of wavelengths.’ The results presented in fig 6 and table 2 could only address statements something like’.the two results appear to be similar within experimental errors..’. Particularly, the preferred technique (BFWA) was well tried and the results were obtained using ‘optimal’ model parameters. How does it perform when sub-optimal model parameters are employed? What about the other methods and are they tested using optimal model parameters too? The authors must address these issues and to make appropriate statements in the abstract and conclusion sections accordingly.
Response 1: Thanks for your comment. We have corrected some inappropriate descriptions in line 22-27, line 386-390, and line 422-426 in the revised manuscript. When comparing with BFWA method, we used the optimal model parameters for GA and other methods through many experiments to ensure the rationality of the experimental results. We have added the necessary description in line 367-369.
Point 2: In my previous review the following questions have not been addressed in this version:
a. ‘section 3: Please check fig 3: is the legend not labelled correctly? How could the 40mins dried sample exhibits a lower norm that that of the 70mins dried spectra? Also, the figure shows a ‘mean’ spectra but the text didn’t mention whether this mean is the average of the spectra over several pixels of the same soybean or it is the average over several soybeans?’
b. ‘Table 2: please add a row of averaged figures (Rp, RMSEP, std etc) for GA vs BFWA over the 10 runs at the bottom of the table?’
c. ‘The conclusion in section 5 may be a bit over interpreted, furthermore, all results presented is based on one sample set of soybeans: it may be better (though not absolute essential) to include other species of soybeans or other crops to strengthen the argument that the BFWA performs more robust than all adopted technique!’
Response 2: Thanks for your comment. We have revised the above questions in the revised manuscript.
a. The labels of Fig. 3 are reasonable. This curve distribution is mainly caused by physical and chemical changes in the drying process, and the details have been added in line 293-304 of the revised manuscript. Five mean reflectance curves in Fig. 3 were obtained by randomly selecting one portion (5 soybeans) and each of them was selected from different drying time soybean samples respectively. Each mean reflectance value curve is the average of the spectra over 5 soybeans. We revised the text in line 292-293.
b. As suggested, we have added a row of averaged figures (Rp, RMSEP, std etc) for GA vs BFWA at the bottom of the table 2.
c. Thanks for your comment. The conclusion section has been corrected in line 426-427 in the revised manuscript.
Point 3: Use of English: there are quite a few grammatical/spelling mistakes throughout the manuscript that are needed to address.
Response 3: Thanks for your comment. We have corrected the grammatical/spelling mistakes in the manuscript.

Reviewer 2 Report
The authors addressed correctly all the comments previously submitted. Only two minor comments:
- Please, review carefully the grammar and typo mistakes in the new text included in the manuscript.
- It would be interesting if authors could include in the discussion section the answer provided to the comment related to the quantitative evaluation of the computational time of the algorithms to reinforce the selection of the 12 bands respect to the full wavelength spectrum.
Author Response
Response to Reviewer 1 Comments
Point 1: Please, review carefully the grammar and typo mistakes in the new text included in the manuscript.
Response 1: Thanks for your comment. We have corrected the grammar and typo mistakes in the manuscript.
Point 2: It would be interesting if authors could include in the discussion section the answer provided to the comment related to the quantitative evaluation of the computational time of the algorithms to reinforce the selection of the 12 bands respect to the full wavelength spectrum.
Response 2: Thanks for your comment. We have added the computational time of soybean moisture content prediction with different number of wavelengths in line 404-407.

Reviewer 3 Report
Summary
This work explores the use of Vis/NIR hyperspectral imaging to measure moisture content during soybean drying using an algorithm for wavelength selection
General comments
The paper is well written and the ideas are well explained throughout the text. Nevertheless, some aspects of the paper should be addressed before publication.
Detailed comments.
· Line 42: In the introduction the authors refer some advantages in the use of NIR for moisture detection, however they do not mention the most important, the high sensitivity of NIR to water.
· Line 49: The authors refer to hyperspectral imaging without referring to the spectral region. Hyperspectral imaging can be used in the visible/UV near/mid/far infrared regions or using for example Raman spectroscopy.
· Line 52: Please define “chemical coloration”.
· Line 93: For the non-experts in soybeans authors should explain the values of maturity and variety presented.
· Line 98-99: how many samples were used for each drying time?
· Line 108: The spectra region between 400 and 1000 nm is in the Vis/NIR region not NIR. The NIR region is between 700 and 2500 nm.
· Line 295/297: the authors state that though the model using the entire spectra has speed limitations due to acquisition and processing. However, they do not explain how they intent to implement online an algorithm such as BFWA. The time dispended to process the algorithm isn’t it higher than to process all the spectra?
· Section 3.2: Are the wavelengths chosen by the three algorithms the same?
· Line 339: The wavelengths chosen by the algorithm are dependent on the sample? Would the model need to be rebuilt to analyze a different type of soybean?
Author Response
Response to Reviewer 2 Comments
Point 1: Line 42: In the introduction the authors refer some advantages in the use of NIR for moisture detection, however they do not mention the most important, the high sensitivity of NIR to water.
Response 1: Thanks for your comment. As suggested, the relevant contents sensitivity of NIR to water have been outlined briefly in line 46-49.
Point 2: Line 49: The authors refer to hyperspectral imaging without referring to the spectral region. Hyperspectral imaging can be used in the visible/UV near/mid/far infrared regions or using for example Raman spectroscopy.
Response 2: Yes, it is true that hyperspectral imaging can be used in the visible/UV near/mid/far infrared regions or using for Raman spectroscopy. However, hyperspectral imaging technique can obtain the spectral and spatial information of samples simultaneously, regardless of the wavelength used. At present, the most used wavelengths of hyperspectral imaging system is 400-1,000nm or 970-1,700nm. In this study, the hyperspectral imaging system covering 400- 1,000 nm wavelengths was used.
Point 3: Line 52: Please define “chemical coloration”.
Response 3: Thanks for your comment. We have revised the manuscript in line 52-54.
Point 4: Line 93: For the non-experts in soybeans authors should explain the values of maturity and variety presented.
Response 4: Thanks for your comment. The values of variety presented (75-3) is the soybean variety code in China. According to Chinese quality standard of vegetable soybean, vegetable soybean maturity can be divided into 10 grades according to the hardness and moisture content. The maturity value 8.5-9 indicate that pods are fully mature and cannot be eaten as vegetable. We believe that the variety and maturity have no effect on the experiment, so we removed these information from the revised manuscript.
Point 5: Line 98-99: how many samples were used for each drying time?
Response 5: Thanks for your comment. About 200g vegetable soybeans were dried by using a pulse spray microwave vacuum drying equipment. The drying times were set to 0-80 min with step of 10 min (here, 0 min means the soybeans are not dried). The dried soybeans were randomly divided into 10 portions (about 5 soybeans for each portion), and each portion was treated as one sample for image collection and moisture measurement. Each drying time experiment was repeated thrice to eliminate unnecessary measurement errors. Hence, a total of 270 (10 samples, 3 repeats, 9 drying time) samples were obtained after drying. We have revised the manuscript in line 98-104.
Point 6: Line 108: The spectra region between 400 and 1000 nm is in the Vis/NIR region not NIR. The NIR region is between 700 and 2500 nm.
Response 6: Yes, we agree with you. The imaging spectrograph used in our experiment is 1003A-10140 Hyperspc™ VNIR C-Series, which covers effective range of 400-1000nm.
Point 7: Line 295/297: the authors state that though the model using the entire spectra has speed limitations due to acquisition and processing. However, they do not explain how they intent to implement online an algorithm such as BFWA. The time dispended to process the algorithm isn’t it higher than to process all the spectra?
Response 7: Thanks for your comment. Hyperspectral image usually contains hundreds of images (each image represents the sample information at special wavelength). It is difficult to meet the need of real-time detection due to high time-cost resulting from collecting, transmitting, storing and processing these images. Hence, developing multispectral imaging system using a small number of important wavelengths is necessary. How to identify important wavelengths from numerous wavelengths to develop multispectral imaging system is particularly important. In this study, we proposed the novel BFWA algorithm to select 12 important wavelengths from 94 wavelengths ranging from 400 to 1000 nm for moisture measurement. These 12 wavelengths can be used to develop multispectral imaging system, which is useful to real-time moisture measurement and quality control during soybean drying process. It should mentioned that the time- consume by collecting, transmitting, and storing image is much greater than those by image processing. Due to the limitations of the experimental conditions, we are unable to provide the time consume by collecting, transmitting and storing image. When all the factors affecting time consume are considered synthetically, the multispectral imaging system developed using a small number of wavelengths selected by BFWA will have great potential in industry application.
Point 8: Section 3.2: Are the wavelengths chosen by the three algorithms the same?
Response 8: Thanks for your comment. The wavelengths selected by three algorithms (SPA, UVE and BFWA) are different, because these algorithms are based different work principle. SPA is a forward selection method and aims to solve the colinearity problem with minimal redundancy. UVE is a method of variable selection based on stability analysis of regression coefficient. The goal of BFWA is to obtain the best accuracy performance by selecting the optimal wavelengths.
Point 9: The wavelengths chosen by the algorithm are dependent on the sample? Would the model need to be rebuilt to analyze a different type of soybean?
Response 9: Thanks for your comment. All of the wavelength selecting method, including UVE, SPA, GA, and BFWA, depend on the sample information. Commonly, we should rebuild or update the model using different wavelengths, if there are big difference among the different type of soybean. How to rebuild or update the model is an important issue in chemometrics, which is beyond the scope of this manuscript.